

# Using application benchmark call graphs to quantify and improve the practical relevance of microbenchmark suites

Martin Grambow[1], Christoph Laaber[2], Philipp Leitner[3] and David Bermbach[1]

[1] Mobile Cloud Computing Research Group, TU Berlin & Einstein Center Digital Future, Berlin, Germany
[2] Department of Informatics, University of Zurich, Zurich, Switzerland
[3] Software Engineering Division, Chalmers | University of Gothenburg, Gothenburg, Sweden

## ABSTRACT

Performance problems in applications should ideally be detected as soon as they occur, i.e., directly when the causing code modification is added to the code repository. To this end, complex and cost-intensive application benchmarks or lightweight but less relevant microbenchmarks can be added to existing build pipelines to ensure performance goals. In this paper, we show how the practical relevance of microbenchmark suites can be improved and verified based on the application flow during an application benchmark run. We propose an approach to determine the overlap of common function calls between application and microbenchmarks, describe a method which identifies redundant microbenchmarks, and present a recommendation algorithm which reveals relevant functions that are not covered by microbenchmarks yet. A microbenchmark suite optimized in this way can easily test all functions determined to be relevant by application benchmarks after every code change, thus, significantly reducing the risk of undetected performance problems. Our evaluation using two time series databases shows that, depending on the specific application scenario, application benchmarks cover different functions of the system under test. Their respective microbenchmark suites cover between 35.62% and 66.29% of the functions called during the application benchmark, offering substantial room for improvement. Through two use cases—removing redundancies in the microbenchmark suite and recommendation of yet uncovered functions—we decrease the total number of microbenchmarks and increase the practical relevance of both suites. Removing redundancies can significantly reduce the number of microbenchmarks (and thus the execution time as well) to ~10% and ~23% of the original microbenchmark suites, whereas recommendation identifies up to 26 and 14 newly, uncovered functions to benchmark to improve the relevance.

By utilizing the differences and synergies of application benchmarks and microbenchmarks, our approach potentially enables effective software performance assurance with performance tests of multiple granularities.

Corresponding author
Martin Grambow,
grambow@tu-berlin.de

## INTRODUCTION

With the continuously increasing complexity of software systems, the interest in reliable and easy-to-use test and evaluation mechanisms has grown as well. While a variety of techniques, such as unit and integration testing, already exists for the validation of functional requirements of an application, mechanisms for ensuring non-functional requirements, e.g., performance, are used more sparingly in practice (*Ameller et al., 2012*; *Caracciolo, Lungu & Nierstrasz, 2014*). Besides live testing techniques such as canary releases (*Schermann, Cito & Leitner, 2018*), developers and researchers usually resort to benchmarking, i.e., the execution of an artificially generated workload against the system under test (SUT), to study and analyze non-functional requirements in artificial production(-near) conditions.

While application benchmarks are the gold standard and very powerful as they benchmark complete systems, they are hardly suitable for regular use in continuous integration pipelines due to their long execution time and high costs (*Bermbach et al., 2017*; *Bermbach & Tai, 2014*). Alternatively, less complex and therefore less costly microbenchmarks could be used, which are also easier to integrate into build pipelines due to their simpler setup (*Laaber & Leitner, 2018*). However, a simple substitution can be dangerous: on the one hand, it is not clear to what extent a microbenchmark suite covers the functions used in production; on the other hand, often only a complex application benchmark is suitable for evaluating complex aspects of a system. To link both benchmark types, we introduce the term *practical relevance* which refers to the extent to which a microbenchmark suite targets code segments that are also invoked by application benchmarks.

In this paper, we aim to determine, quantify, and improve the practical relevance of a microbenchmark suite by using application benchmarks as a baseline. In real setups, developers often do not have access to a (representative) live system, e.g., generally-available software such as database systems are used by many companies which install and deploy their own instances and, consequently, the software's developers usually do not have access to the custom installations and their production traces and logs. In addition, software is used differently by each customer which results in different load profiles as well as varying configurations. Thus, it is often reasonable to use one or more application benchmarks as the next accurate proxy to simulate and evaluate a representative artificial production system. The execution of these benchmarks for each code change is very expensive and time-consuming, but a light-weight microbenchmark suite that has proven to be practically relevant could replace them to some degree.

To this end, we analyze the called functions of a reference run, which can be (an excerpt from) a production system or an application benchmark, and compare them with the functions invoked by microbenchmarks to determine and quantify a microbenchmark suite's practical relevance. If every called function of the reference run is also invoked by at least one microbenchmark, we consider the respective microbenchmark suite as 100% practically relevant as the suite covers all functions used in the baseline execution. Based on this information, we devise two optimization strategies to improve the practical relevance

of microbenchmark suites according to a reference run: (i) a heuristic for removing redundancies in an existing microbenchmark suite and (ii) a recommendation algorithm which identifies uncovered but relevant functions.

In this regard, we formulate the following research questions:

**RQ1** How to determine and quantify the practical relevance of microbenchmark suites?

Software source code in an object-oriented system is organized in classes and functions. At runtime, executed functions call other classes and functions, which leads to a program flow that can be depicted as a call graph. This graph represents which functions call which other functions and adds additional meta information such as the duration of the executed function. If these graphs are available for a reference run and the respective microbenchmark suite, it is possible to compare the flow of both graphs and quantify to which degree the current microbenchmark suite reflects the use in the reference run, or rather the real usage in production. Our evaluation with two well-known time series databases shows that their microbenchmark suites cover about 40% of the functions called during application benchmarks. The majority of the functionality used by an application benchmark, our proxy for a production application, is therefore uncovered by the microbenchmark suites of our study objects.

**RQ2** How to reduce the execution runtime of microbenchmark suites without affecting their practical relevance?

If there are many microbenchmarks in a suite, they are likely to have redundancies and some functions will be benchmarked by multiple microbenchmarks. By creating a new subset of the respective microbenchmark suite without these redundancies, it is possible to achieve the same coverage level with fewer microbenchmarks, which significantly reduces the overall runtime of the microbenchmark suite. Applying this optimization as part of our evaluation shows that this can reduce the number of microbenchmarks by 77% to 90%, depending on the application and benchmark scenario.

**RQ3** How to increase the practical relevance within cost efficiency constraints?

If the microbenchmark suite's coverage is not sufficient, the uncovered graph of the application benchmark can be used to locate functions which are highly relevant for practical usage. We present a recommendation algorithm which provides a fast and automated way to identify these functions that should be covered by microbenchmarks. Our evaluation shows that an increase in coverage from the original 40% to up to 90% with only three additional microbenchmarks is theoretically possible. An optimized microbenchmark suite could, e.g., serve as initial and fast performance smoke test in continuous integration or deployment (CI/CD) pipelines or for developers who need a quick performance feedback for their recent changes.

After applying both optimizations, it is possible to cover a maximum portion of an application benchmark with a minimum suite of microbenchmarks which has several advantages. First of all, this helps to identify important functions that are relevant in practice and ensures that their performance is regularly evaluated via microbenchmarks. Instead of a suite that checks rarely used functions, code sections that are relevant for practical use are evaluated frequently. Second, microbenchmarks evaluating functions that are already implicitly covered by other microbenchmarks are selectively removed,

achieving the same practical relevance with as few microbenchmarks as possible while reducing the runtime of the total suite. Furthermore, the effort for the creation of microbenchmarks is minimized because the microbenchmarks of the proposed functions will cover a large part of the application benchmark call graph and fewer microbenchmarks are necessary. Developers will still have to design and implement performance tests, but the identification of highly relevant functions for actual operation is facilitated and functions that implicitly benchmark many further relevant functions are pointed out, thus covering a broad call graph. Ultimately, the optimized microbenchmark suite can be used in CI/CD pipelines more effectively: It is possible to establish a CI/CD pipeline which, e.g., executes the comparatively simple and short but representative microbenchmark suite after each change in the code. The complex and cost-intensive application benchmark can then be executed more sparsely, e.g., for each major release. In this sense, the application benchmark remains as the gold standard revealing all performance problems, while the optimized microbenchmark suite is an easy-to-use and fast heuristic which offers a quick insight into performance yet with obviously lower accuracy.

It is our hope that this study contributes to the problem of performance testing as part of CI/CD pipelines and enables a more frequent validation of performance metrics to detect regressions sooner. Our approach can give targeted advice to developers to improve the effectiveness and relevance of their microbenchmark suite. Throughout the rest of the paper, we will always use an application benchmark as the reference run but our approach can, of course, also use other sources as a baseline.

**Contributions:**
- An approach to determine and quantify the practical relevance of a microbenchmark suite.
- An adaptation of the Greedy-based algorithm proposed by *Chen & Lau (1998)* to remove redundancies in a microbenchmark suite.
- A recommendation strategy inspired by *Rothermel et al. (1999)* for new microbenchmarks which aims to cover large parts of the application benchmark's function call graph.
- An empirical evaluation which analyzes and applies the two optimizations to the microbenchmark suites of two large open-source time series databases.

**Paper Structure:** After summarizing relevant background information in "Background", we present our approach to determine, quantify, and improve microbenchmark suites in "Approach". Next, we evaluate our approach by applying the proposed algorithms to two open-source time series databases in "Empirical Evaluation" and discuss its strength and limitations in "Discussion". Finally, we outline related work in "Related Work" and conclude in "Conclusion".

## BACKGROUND

This section introduces related background information, in particular this comprises benchmarking and time series databases.

## Benchmarking

Benchmarking aims to determine quality of service (QoS) by stressing a system under test (SUT) in a standardized way while observing its reactions, usually in a test or staging environment (*Bermbach et al., 2017*; *Bermbach, Wittern & Tai, 2017*). To provide relevant results, benchmarks must meet certain requirements such as fairness, portability, or repeatability (*Huppler, 2009*; *Bermbach et al., 2017*; *Bermbach, Wittern & Tai, 2017*; *Folkerts et al., 2013*). In this paper, we deal with two different kinds of benchmarks: application benchmarks, which evaluate a complete application system, and microbenchmarks, which evaluate individual functions or methods. Functional testing as well as monitoring are not a focus of this work, but are of course closely related (*Bermbach, Wittern & Tai, 2017*).

### Application benchmark

In a so-called application benchmark, the SUT is first set up and brought into a defined initial state, e.g., using warmup requests or by inserting initial data. Next, an evaluation workload is sent to the SUT and the relevant metrics are collected. This method is on the one hand very powerful, because many relevant aspects and conditions can be simulated in a defined testbed, but it is very expensive and time-consuming on the other hand; not only in the preparation but also in the periodic execution.

The evaluation of an entire system involves several crucial tasks to finally come up with a relevant comparison and conclusion, especially in dynamic cloud environments (*Bermbach et al., 2017*; *Bermbach, Wittern & Tai, 2017*; *Grambow, Lehmann & Bermbach, 2019*). During the design phase, it is necessary to think in detail about the specific requirements of the benchmark and its objectives. While defining (and generating) the workload, many aspects must be taken into account to ensure that the requirements of the benchmark are not violated and to guarantee a relevant result later on (*Huppler, 2009*; *Bermbach et al., 2017*; *Bermbach, Wittern & Tai, 2017*; *Folkerts et al., 2013*). This is especially difficult in dynamic cloud environments, because it is hard to reproduce results due to performance variations inherent in cloud systems, random fluctuations, and other cloud-specific characteristics (*Lenk et al., 2011*; *Difallah et al., 2013*; *Folkerts et al., 2013*; *Rabl et al., 2010*). To set up an SUT, all components have to be defined and initialized first. This can be done with the assistance of automation tools (e.g., *Hasenburg et al., 2019*; *Hasenburg, Grambow & Bermbach, 2020*). However, automation tools still have to be configured first, which further complicates the setup of application benchmarks. During the benchmark run, all components have to be monitored to ensure that there is no bottleneck inside the benchmarking system, e.g., to avoid quantifying the resources of the benchmarking client's machine instead of the maximum throughput of the SUT. Finally, the collected data needs to be transformed into relevant insights, usually in a subsequent offline analysis (*Bermbach, Wittern & Tai, 2017*). Together, these factors imply that a really *continuous* application benchmarking, e.g., applied to every code change, will usually be prohibitively expensive in terms of time but also in monetary cost.

### Microbenchmarks

Instead of benchmarking the entire SUT at once, microbenchmarks focus on benchmarking small code fragments, e.g., single functions. Here, only individual critical or often used functions are benchmarked on a smaller scale (hundreds of invocations) to ensure that there is no performance drop introduced with a code change or to estimate rough function-level metrics, e.g., average execution duration or throughput. They are usually defined in only a few lines of code; while they are also executed repeatedly, running microbenchmarks takes considerably less time than the execution of an application benchmark. Moreover, they are usually easier to set up and to execute as there is no complex SUT which needs to be initialized first. They are therefore more suitable for frequent use in CI/CD pipelines but also have to cope with variability in cloud environments (*Leitner & Bezemer, 2017*; *Laaber & Leitner, 2018*; *Laaber, Scheuner & Leitner, 2019*; *Bezemer et al., 2019*). Finally, they cannot cover all aspects of an application benchmark and are, depending on the concrete use-case, usually considered less relevant individually.

## Time series database systems

In this paper, we use time series database systems (TSDBs) as study objects. TSDBs are designed and optimized to receive, store, manage, and analyze time series data (*Dunning et al., 2014*). Time series data usually comprises sequences of timestamped data—often numeric values—such as measurement values. As these values tend to arrive in-order, TSDB storage layers are optimized for append-only writes because only a few straggler values arrive late, e.g., due to network delays. Moreover, the stored values are rarely updated as the main purpose of TSDBs is to identify trends or anomalies in incoming data, e.g., for identifying failure situations. Due to this, TSDBs are optimized for fast aggregation queries over variable-length time frames. Furthermore, most TSDBs support tagging which is needed for grouping values by dimension in queries. Taken together, these features and performance-critical operations make TSDBs a suitable study object for the evaluation of our approach. Examples of TSDBs include InfluxDB (https://www.influxdata.com), VictoriaMetrics (https://victoriametrics.com), Prometheus (https://prometheus.io), and OpenTSDB (http://opentsdb.net).

## APPROACH

We aim to determine and quantify the practical relevance of microbenchmark suites, i.e., to what extent the functions invoked by application benchmarks are also covered by microbenchmarks. Moreover, we want to improve microbenchmark suites by identifying and removing redundancies as well as recommending important functions which are not covered yet.

Our basic idea is based on the intuition that, regardless of whether software is evaluated by an application benchmark or microbenchmark, both types evaluate the same source code and algorithms. Since an application benchmark is designed to simulate realistic operations in a production-near environment, it can reasonably be assumed that it can serve as a baseline or reference execution to quantify relevance in the absence of a real

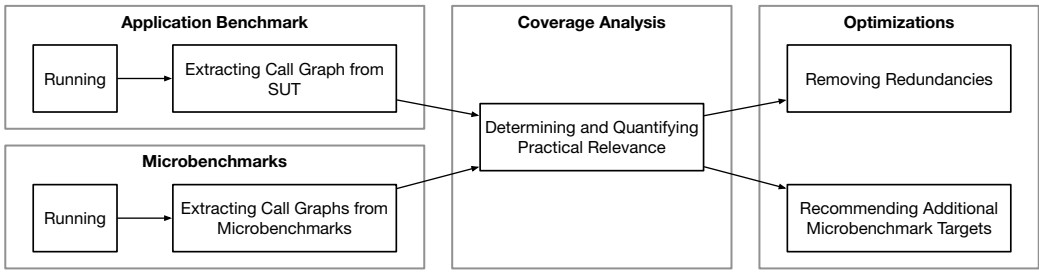

**Figure 1** A study subject (system) is evaluated via application benchmark and its microbenchmark suite, the generated call graphs during the benchmark runs are compared to determine and quantify the practical relevance, and two use cases to optimize the microbenchmark suite are proposed.

production trace. On the other hand, microbenchmarks are written to check the performance of individual functions and multiple microbenchmarks are bundled as a microbenchmark suite to analyze the performance of a software system. Both benchmark types run against the same source code and generate a program flow (call graph) with functions[1] as nodes and function calls as edges. We propose to analyze these graphs to (i) determine the coverage of both types to quantify the practical relevance of a microbenchmark suite, (ii) remove redundancies by identifying functions (call graph nodes) which are covered by multiple microbenchmarks, and (iii) recommend functions which should be covered by microbenchmarks because of their usage in the application benchmark. In a perfectly benchmarked software project, the ideal situation in terms of our approach would be that all practically relevant functions are covered by exactly one microbenchmark. To check and quantify this fact for a given project and to improve it subsequently, we propose the approach illustrated in Fig. 1.

To use our approach, we assume that the software project complies with best practices for both benchmarking domains, e.g., *Bermbach, Wittern & Tai (2017)*, *Damasceno Costa et al. (2019)*. It is necessary that there is both a suite of microbenchmarks and at least one application benchmark for the respective SUT. The application benchmark must rely on realistic scenarios to generate a relevant program flow and must run against an instrumented SUT which can create a call graph during the benchmark execution. During that tracing run, actual measurements of the application benchmark do not matter. The same applies for the execution of the microbenchmarks, where it also must be possible to reliably create the call graphs for the duration of the benchmark run. These call graphs can subsequently be analyzed structurally to quantify and improve the microbenchmark suite's relevance. We will discuss this in more detail in "Determining and Quantifying Relevance".

We propose two concrete methods for optimizing a microbenchmark suite: (1) An algorithm to remove redundancies in the suite by creating a minimal sub-set of microbenchmarks which structurally covers the application benchmark graph to the same extent (see "Determining and Quantifying Relevance"). (2) A recommendation strategy to suggest individual functions which are currently not covered by microbenchmarks but

[1] In the following, we exclusively refer to functions but our approach can similarly be used for methods and procedures depending on the SUT's programming language.

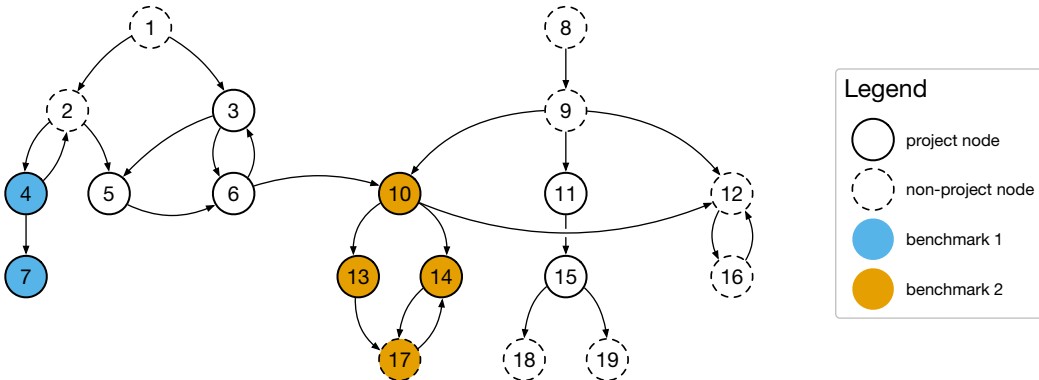

**Figure 2** The practical relevance of a microbenchmark suite can be quantified by by relating the number of covered functions and the total number of called functions during an application benchmark to each other.     

which are relevant for the application benchmark and will cover a large part of its call graph (see "Recommending Additional Microbenchmark Targets").

## Determining and quantifying relevance

After executing an application benchmark and the microbenchmark suite on an instrumented SUT, we retrieve one (potentially large) call graph from the application benchmark run and many (potentially small) graphs from the microbenchmark runs, one for each microbenchmark. In these graphs, each function represents a node and each edge represents a function call. Furthermore, we differentiate in the graphs between so-called project nodes, which refer directly to functions of the SUT, and non-project nodes, which represent functions from libraries or the operating system. After all graphs have been generated, the next step is to determine the function coverage, i.e., which functions are called by both the application benchmark and at least one microbenchmark.

Figure 2 shows an example: The application benchmark graph covers all nodes from node 1 to node 19 and has two entry points, node 1 and node 8. These entry points, when invoked, call other functions, which again call other functions (cycles are possible, e.g., in the case of recursion). Nodes in the graph can be both project functions of the SUT or functions of external libraries. There are also two microbenchmarks in this simple example, benchmark 1 and benchmark 2. While benchmark 1 only covers two nodes, benchmark 2 covers four nodes and seems to be more practically relevant (we will discuss this later in more detail).

To determine the function coverage, we iterate through all application benchmark nodes and identify all microbenchmarks which cover this function. As a result, we get a list of coverage sets, one for each microbenchmark, where each entry describes the overlap of nodes (functions) between the application benchmark call graph and the respective microbenchmark graph. Next, we count (i) all project-only functions and (ii) all functions which are called during the application benchmark and in at least one microbenchmark. Finally, we calculate two different coverage metrics: First, the *project-only* coverage of all

---

**Algorithm 1** Removing redundancies in the microbenchmark suite.

---

**Input:** *C*- Coverage sets

**Result:** *minimalSet* - Minimal set of microbenchmarks

1 *minimalSet* ← ø

2 **while** |C| > 0 **do**

3    *C* ← Sortsets (*C*)

4    *largestCoverage* ← RemoveFirst (*C*)

5    **if** |*largestCoverage*| == 0 **then**

6        **return** *minimalSet*

7    **end**

8    *minimalSet* ← *minimalSet* ∪ *largestCoverage*

9    **foreach** set ∈ *C* **do**

10       *set* ← *set* \ *largestCoverage*

11    **end**

12 **end**

---

executed functions in comparison to the total number of project functions in the application benchmark. Second, the *overall* coverage, including external functions.

For our example application benchmark call graph in Fig. 2: $coverage_{project-only} = \frac{5}{10} = 0.5$ and $coverage_{overall} = \frac{6}{19} \approx 0.316$. Note that these metrics would not change if there would be a third microbenchmark covering a subset of already covered nodes, e.g., node 14 and node 17.

## Removing redundancies

Our first proposed optimization removes redundancies in the microbenchmark suite and achieves the same coverage level with fewer microbenchmarks. For example, the imaginary third benchmark mentioned above (covering nodes 14 and 17 in Fig. 2) would be redundant, as all nodes are already covered by other microbenchmarks. To identify a minimal set of microbenchmarks, we adapt the Greedy algorithm proposed by *Chen & Lau (1998)* and rank the microbenchmarks based on the number of reachable function nodes that overlap with the application benchmark (instead of *all* reachable nodes as proposed in *Chen & Lau (1998)*), as defined in Algorithm 1.

After analyzing the graphs, we get coverage sets of overlaps between the application benchmark and the microbenchmark call graphs (input *C*). First, we sort them based on the number of covered nodes in descending order, i.e., microbenchmarks which cover many functions of the application benchmark are moved to the top (line 3). Next, we pick the first coverage set as it covers the most functions and add the respective microbenchmark to the minimal set (lines 4 to 8). Afterwards, we have to remove the covered set of the selected microbenchmark from all coverage sets (lines 9 to 11) and sort the coverage set again to pick the next microbenchmark. We repeat this until there are no more microbenchmarks to add (i.e., all microbenchmarks are part of the minimal set and

there is no redundancy) or until the picked coverage set would not add any covered functions to the minimal set (line 6).

In this work, we sort the coverage sets by their number of covered nodes and do not include any additional criteria to break ties. This could, however, result in an undefined outcome if there are multiple coverage sets with the same number of covered additional functions, but this case is a rare event and did not occur in our study. Still, including other secondary sorting criteria such as the distance to the graph's root node or the total number of nodes in the coverage set might improve this optimization further.

## Recommending additional microbenchmark targets

A well-designed application benchmark will trigger the same function calls in an SUT as a production use would. A well-designed microbenchmark for an individual function will also implicitly call the same functions as in production or during the application benchmark. In this second optimization of the microbenchmark suite, we rely on these assumptions to selectively recommend uncovered functions for further microbenchmarking. This allows developers to directly implement new microbenchmarks that will cover a large part of the uncovered application benchmark call graph and thus increase the coverage levels (see "Determining and Quantifying Relevance").

Similar to the removal of redundancies, we build on the idea of a well-known, greedy test case prioritization algorithm proposed by *Rothermel et al. (1999)* to recommend functions for benchmarking that are not covered yet. In particular, we adapt Rothermel's *additional algorithm*, which iteratively prioritizes tests whose coverage of new parts of the program (that have not been covered by previously prioritized tests) is maximal. Instead of using the set of all covered methods by a microbenchmark suite, our adaptation uses the function nodes from the application benchmark that are not covered yet as greedy criteria to optimize for.

Algorithm 2 defines the recommendation algorithm. The algorithm requires as input the call graph from the application benchmark, the graphs from the microbenchmark suite, and the coverage sets determined in "Determining and Quantifying Relevance", as well as the upper limit $n$ of recommended functions.

First, we determine the set of nodes (functions) in the application benchmark call graph which are not covered by any microbenchmark (line 2). Next, we determine the reachable nodes for each function in this set, only considering project nodes, and store the results in another set $N$ (lines 5 to 7). To link back to our example graph in Fig. 2, the resulting set for function 3 (neither covered by benchmark 1 nor 2) would be nodes 3, 5, and 6 (node 12 is not a project node and not part of the reachable nodes). Third, we sort the set $N$ by the number of nodes in each element, starting with the set with the most nodes in it (line 8). If two functions cover the same number of project nodes, we determine the distance to the closest root node and select functions that have a shorter distance. If the functions are still equivalent, we include the number of covered non-project nodes as a third factor and favor the function with higher coverage. Finally, we pick the first element and add the respective function to the recommendation set $R$ (lines 9 to 13), update the not covered functions (line 15), and run the algorithm again to find the next

---

**Algorithm 2** Recommending functions which are not covered by microbenchmarks yet.

**Input**: $\langle A, M, C \rangle$ - Application benchmark CG, microbenchmark CGs, coverage sets

**Output**: $n$ - Number of microbenchmarks to recommend

**Result**: $R$ - Set of recommended functions to microbenchmark

1  $R \leftarrow \varnothing$

2  $notCovered \leftarrow \{a | a \in A \wedge \texttt{IsProjectNode}(a)\} \setminus C^{total}$

3  $N \leftarrow \varnothing$

4  **while** $n > 0$ **do**

5    **foreach** *function* $f_a \in notCovered$ **do**

6      $additionalNodes \leftarrow \texttt{DetermineReachableNodes}(f_a) \cap notCovered$

        $N \leftarrow N \cup \{additionalNodes\}$

7    **end**

8    $\texttt{SortByNumberOfNodes}(N)$

9    $largestAdditionalSet \leftarrow \texttt{RemoveFirst}(N)$

10    **if** $|largestAdditionalSet| == 0$ **then**

11      **return** $R$

12    **end**

13    $R \leftarrow R \cup largestAdditionalSet[0]$

14    $n = n - 1$

15    $notCovered \leftarrow notCovered \setminus largestAdditionalSet$

16  **end**

---

function which adds the most additional nodes to the covered set. Our algorithm ends if there are $n$ functions in $R$ (i.e., upper limit for recommendations reached) or if the function which would be added to the recommendation set $R$ does not add additional functions to the covered set (line 11).

# EMPIRICAL EVALUATION

We empirically evaluate our approach on two open-source TSDBs written in Go, namely *InfluxDB* and *VictoriaMetrics*, which both have extensive developer-written microbenchmark suites. As application benchmark and, therefore, baseline, we encode three application scenarios in YCSB-TS (https://github.com/TSDBBench/YCSB-TS). On the other side, we run the custom microbenchmark suites of the respective systems.

We start by giving an overview of YCSB-TS and both evaluated systems in "Study Objects". Next, we describe how we run the application benchmark using three different scenarios in "Application Benchmark" and the microbenchmark suite in "Microbenchmarks" to collect the respective set of call graphs. Finally, we use the call graphs to determine the coverage for each application scenario and quantify the practical relevance in "Determining and Quantifying Relevance" before removing redundancies in

**Table 1 Our evaluation uses two open-source TSDBs written in Go as study objects.**

| Project | InfluxDB | VictoriaMetrics |
|---|---|---|
| GitHub URL | influxdata/influxdb | VictoriaMetrics/VictoriaMetrics |
| Branch/Release | 1.7 | v1.29.4 |
| Commit | ff383cd | 2ab4cea |
| Go Files | 646 | 1,284 |
| Lines Of Code (Go) | 193,225 | 462,232 |
| Contributers | 407 | 32 |
| Stars | ca. 19,100 | 2,500 |
| Forks | ca. 2,700 | 185 |
| Microbenchmarks in Project | 347 | 65 |
| Extracted Call Graphs | 288 | 62 |

the benchmark suites in "Removing Redundancies" and recommending functions which should be covered by microbenchmarks for every investigated project in "Recommending Additional Microbenchmark Targets".

## Study objects

To evaluate our approach, we need an SUT which comes with a developer-written microbenchmark suite *and* which is compatible with an application benchmark. For this, we particularly looked at TSDBs written in Go as they are compatible with the YCSB-TS application benchmark, and since Go contains a microbenchmark framework as part of its standard library. Furthermore, Go provides a tool called pprof (https://golang.org/pkg/runtime/pprof) which allows us to extract the call graphs of an application using instrumentation. Based on these considerations, we decided to evaluate our approach with the TSDBs *InfluxDB* (https://www.influxdata.com) and *VictoriaMetrics* (https://victoriametrics.com) (see Table 1).

YCSB-TS (https://github.com/TSDBBench/YCSB-TS) is a specialized fork of YCSB (*Cooper et al., 2010*), which is an extensible benchmarking framework for data serving systems, for time series databases. Usually every experiment with YCSB is divided into a load phase which preloads the SUT with initial data, and a run phase which executes the actual experiment queries.

*InfluxDB* is a popular TSDB with more than 400 contributors and more than 19,000 stars on GitHub. *VictoriaMetrics* is an emerging TSDB (the first version was released in 2018) which has already collected more than 2,000 stars on GitHub. Both systems are written in Go, offer a microbenchmark suite, and can be benchmarked using the YCSB-TS tool. However, there was no suitable connector for *VictoriaMetrics* in the official YCSB-TS repository; we therefore implemented one based on the existing connectors for *InfluxDB* and Prometheus. Moreover, we also fixed some small issues in the YCSB-TS implementation. A fork with all necessary changes, including the new connector and all fixes, is available on GitHub (https://github.com/martingrambow/YCSB-TS).

**Table 2 We configured an application benchmark to use three different workload profiles.**

| Scenario | Medical monitoring | Smart factory | Wind parks |
|---|---|---|---|
| **Load** | | | |
| Records | 1,512,000 | 1,339,200 | 2,190,000 |
| **Run** | | | |
| Insert | 1,512,000 | 1,339,200 | 2,190,000 |
| Scan | 1,680 | 1,860 | 35,040 |
| Avg | 100,800 | 744 | 35,040 |
| Count | 0 | 744 | 0 |
| Sum | 0 | 2,976 | 8,760 |
| Total | 1,614,480 | 1,345,524 | 2,268,840 |
| **Other** | | | |
| Duration | 7 days | 31 days | 365 days |
| Tags | 10 | 10 | 5 |

## Application benchmark

Systems such as our studied TSDBs are used in different domains and contexts, resulting in different load profiles depending on the specific use case. We evaluated each TSDB in three different scenarios which are motivated in "Scenarios". The actual benchmark experiment is described in "Experiments".

### Scenarios

Depending on the workload, the call graphs within an SUT may vary. To consider this effect in our evaluation, we generate three different workloads based on the following three scenarios for TSDBs, see Table 2. All workload files are available on GitHub (https://github.com/martingrambow/YCSB-TS/tree/master/workloads).

**Medical Monitoring:** An intensive care unit monitors its patients through several sensors which forward the tagged and timestamped measurements to the TSDB. These values are requested and processed by an analyzer, which averages relevant values for each patient once per minute and scans for irregularities once per hour. In our workload configuration, we assume a new data item for every patient every two seconds and deal with 10 patients.

We convert this abstract scenario description into the following YCSB-TS workload: With an evaluation period of seven days, there are approximately three million values in the range of 0 to 300 that are inserted into the database in total. Half of them, about 1.5 million, are initially inserted during the load phase. Next, in the run phase, the remaining records are inserted and the queries are made. In this scenario, there are about 100,000 queries which contain mostly AVG as well as 1,680 SCAN operations. Furthermore, the workload uses ten different tags to simulate different patients.

**Smart Factory:** In this scenario, a smart factory produces several goods with multiple machines. Whenever an item is finished, the machine controller submits the idle time during the manufacturing process as a timestamped entry to the TSDB tagged with the kind of produced item. Furthermore, a monitoring tool queries the average and the total

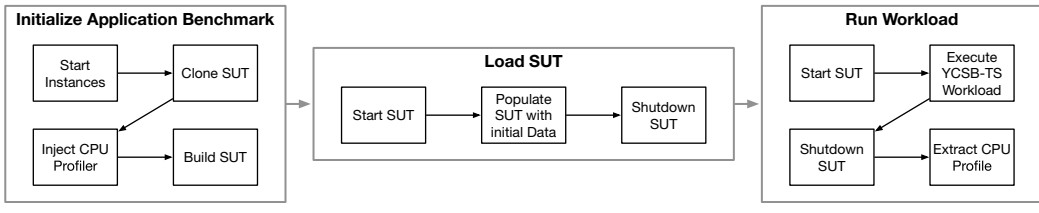

**Figure 3 After initialization, the SUT is filled with initial data and restarted for the actual experiment run to clearly separate the program flow.**

amount of produced items once per hour and the accumulated idle time at each quarter of an hour. Finally, there are several manual SCAN queries for produced items over a given period. Our evaluation scenario deals with five different products and ten machines which on average each assemble a new item every ten seconds. Moreover, there are 60 SCAN queries on average per day.

The corresponding YCSB-TS workload covers a 31-day evaluation period during which approximately 2.6 million data records are inserted. Again, we split the records in half and insert the first part in the load phase and insert the second part in parallel to all other queries in the run phase. In this scenario, we execute about 6,000 queries in the run phase which include SUM, SCAN, AVG, and COUNT operations (frequency in descending order). Furthermore, the workload uses five predefined tags to simulate the different products.

**Wind Parks:** Wind wheels in a wind park send information about their generated energy as timestamped and tagged items to the TSDB once per hour. At each quarter of an hour, a control center scans and counts the incoming data from 500 wind wheels in five different geographic regions. Moreover, it totals the produced energy for every hour.

Translated into a YCSB-TS workload with 365 days evaluation time, this means about 4.4 million records to be inserted and five predefined tags for the respective regions. Again, we have also split the records equally between the load and run phase. In addition, we run about 80,000 queries, split between SCAN, AVG, and SUM (frequency in descending order).

### Experiments

Similar to *Bermbach et al. (2017)*, each experiment is divided into three phases: initialize, load, and run (see Fig. 3). During the initialization phase, we create two AWS t2.medium EC2 instances (2 vCPUs, 4 GiB RAM), one for the SUT and one for the benchmarking client in the eu-west-1 region. The setup of the client is identical for all experiments: YCSB-TS is installed and configured on the benchmarking client instance. The initialization of the SUT starts with the installation of required software, e.g., Git, Go, and Docker. Next, we clone the SUT, revert to a fixed Git commit (see Table 1) and instrument the source code to start the CPU profiling when running. Finally, we build the SUT and create an executable file.

During the load phase, we start the SUT and execute the load workload of the respective scenario using the benchmarking client and preload the database. Then, we stop the

SUT and keep the inserted data. This way we can clearly separate the call graphs of the following run phase from the rest of the experiment.

Afterwards, we restart the SUT for the run phase. Since the source code has been instrumented, a CPU profile is now created and function calls are recorded in it by sampling while the SUT runs. Next, we run the actual workload against the SUT using the benchmarking client and subsequently stop the SUT. This run phase of the experiments took between 40 minutes and 18 hours, depending on the workload and TSDB. Note that the actual benchmark runtime is in this case irrelevant (as long as it is sufficiently long) since we are only interested in the call graph. Finally, we export the generated CPU profile which we use to build the call graphs.

After running the application benchmark for all scenarios and TSDBs, we have six application benchmark call graphs, one for each combination of scenario and TSDB.

## Microbenchmarks

To generate the call graphs for all microbenchmarks, we execute all microbenchmarks in both projects one after the other and extract the CPU profile for each microbenchmark separately. Moreover, we set the benchmark execution time to ten seconds to reduce the likelihood that the profiler misses nodes (functions), due to statistical sampling of stack frames. This means that each microbenchmark is executed multiple times until the total runtime for this microbenchmark reaches ten seconds and that the runtime is usually slightly higher than ten seconds (the last execution starts before the ten seconds deadline and ends afterwards). Finally, we transform the profile files of each microbenchmark into call graphs, which we use in our further analysis.

## Determining and quantifying relevance

Based on the call graphs for all scenario workloads and microbenchmarks, we analyze the coverage of both to determine and quantify the practical relevance following "Determining and Quantifying Relevance". Figure 4 shows the microbenchmark suite's coverage for each study object and scenario. For *InfluxDB*, the overall coverage ranges from 62.90% to 66.29% and the project-only coverage ranges from 40.43% to 41.25%, depending on the application scenario. For *VictoriaMetrics*, the overall coverage ranges from 43.5% to 46.74% and the project-only coverage from 35.62% to 40.43%. Table 3 shows the detailed coverage levels.

As a next step, we also analyze the coverage sets of all application benchmark call graphs to evaluate to which degree the scenarios vary and generate different call graphs. Figures 5A and 5B show the application scenario coverage as Venn diagrams for *InfluxDB* and *VictoriaMetrics* using project nodes only. Both diagrams show the same characteristics in general. All scenarios trigger unique functions which are not covered by other scenarios, see Table 2. For both SUTs, the Smart Factory scenario generates the smallest unique set of project-only nodes (29 unique functions for *InfluxDB* and 4 unique ones for *VictoriaMetrics* ) and the Wind Parks scenario the largest one (134 functions for *InfluxDB* and 77 for *VictoriaMetrics* ). Furthermore, all scenarios also generate a set of common functions which are invoked in every scenario. For *InfluxDB*, there are 464

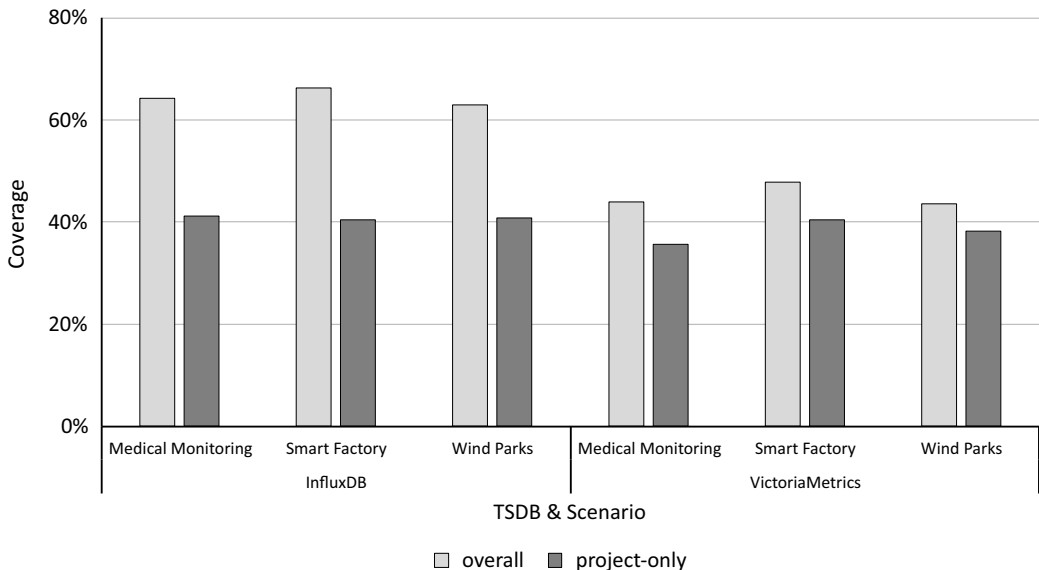

**Figure 4 The project-only coverage is about 40% for both microbenchmark suites, leaving a lot potential room for improvement.**

**Table 3 All microbenchmarks together form a significantly larger call graph than the application benchmark (number of nodes); however, these by far do not cover all functions called during the application benchmarks (coverage).**

| Project | Scenario | Node type | Number of nodes | | Coverage | |
|---|---|---|---|---|---|---|
| | | | **App** | **Micro** | **Abs.** | **Rel. (%)** |
| InfluxDB | Medical monitoring | Overall | 1,838 | 3,069 | 1,180 | 64.20 |
| | | Project-only | 737 | 1,621 | 304 | 41.25 |
| | Smart factory | Overall | 1,504 | 3,069 | 997 | 66.29 |
| | | Project-only | 517 | 1,621 | 209 | 40.43 |
| | Wind parks | Overall | 1,895 | 3,069 | 1,192 | 62.90 |
| | | Project-only | 778 | 1,621 | 318 | 40.87 |
| VictoriaMetrics | Medical monitoring | Overall | 1,573 | 1,125 | 691 | 43.93 |
| | | Project-only | 511 | 454 | 182 | 35.62 |
| | Smart factory | Overall | 1,238 | 1,125 | 591 | 47.74 |
| | | Project-only | 371 | 454 | 150 | 40.43 |
| | Wind parks | Overall | 1,600 | 1,125 | 696 | 43.50 |
| | | Project-only | 542 | 454 | 207 | 38.19 |

functions of 920 in total (50.43%) and for *VictoriaMetrics* there are 341 functions of 603 in total (56.55%) which are called in every application scenario. Table 4 shows the overlap details.

## Removing redundancies

Our first optimization, as defined in Algorithm 1, analyzes the existing coverage sets and removes redundancy from both microbenchmark suites by greedily adding

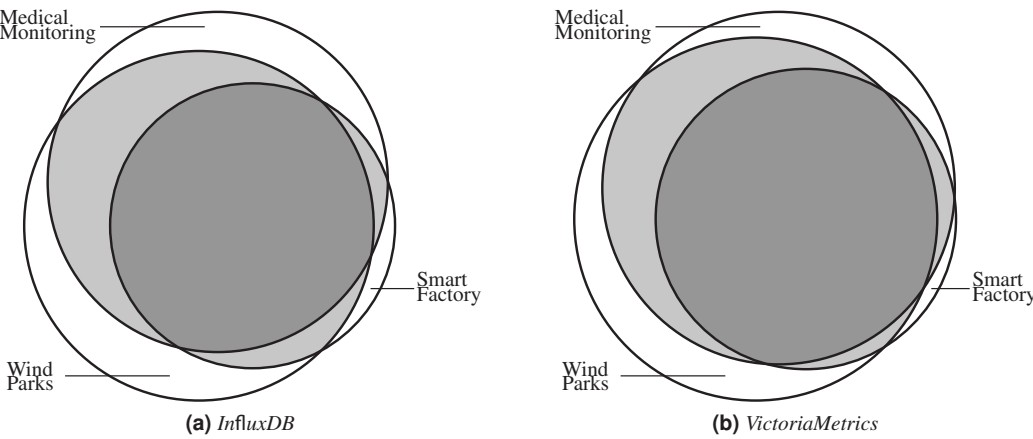

**(a)** *InfluxDB*                **(b)** *VictoriaMetrics*

**Figure 5 All scenarios generate individual call graphs for both SUTs *InfluxDB* (A) and *VictoriaMetrics* (B). Some functions are exclusively called in one scenario, many are called in two or all three scenarios.**

**Table 4 Pair-wise overlap between different scenarios.**

| Project | Node type | Scenario | Medical monitoring | Smart factory | Wind parks |
|---|---|---|---|---|---|
| InfluxDB | Overall | Medical monitoring | same | 1,411 (76.77%) | 1,662 (90.42%) |
| | | Smart factory | 1,411 (93.82%) | same | 1,445 (96.08%) |
| | | Wind parks | 1,662 (87.70%) | 1,445 (76.25%) | same |
| | Project-only | Medical monitoring | same | 468 (63.50%) | 624 (84.67%) |
| | | Smart factory | 468 (90.52%) | same | 484 (93.62%) |
| | | Wind parks | 624 (80.21%) | 484 (62.21%) | same |
| VictoriaMetrics | Overall | Medical monitoring | same | 1,171 (74.4%) | 1,391 (88.43%) |
| | | Smart factory | 1,171 (94.59%) | same | 1,158 (93.54%) |
| | | Wind parks | 1,391 (86.94%) | 1,158 (72.37%) | same |
| | Project-only | Medical monitoring | same | 356 (69.67%) | 454 (88.84%) |
| | | Smart factory | 356 (95.96%) | same | 352 (94.88%) |
| | | Wind parks | 454 (83.76%) | 352 (64.94%) | same |

microbenchmarks to a minimal suite which fulfills the same coverage criteria. Figure 6 shows the step-by-step construction of this minimal set of microbenchmarks up to the maximum possible coverage.

For *InfluxDB* (Fig. 6A), the first selected microbenchmark already covers more than 12% of each application benchmark scenario graph. Furthermore, the first four selected microbenchmarks are identical in all scenarios. Depending on the scenario, these already cover a total of 28% to 31% (with a maximum coverage of about 40% when using all microbenchmarks, see Table 3). These four microbenchmarks are therefore very important when covering a large practically relevant area in the source code. However, even if all microbenchmarks selected during minimization are chosen and the maximum possible coverage is achieved, the removal of redundancies remains very effective. Depending on the application scenario, the initial suite with 288 microbenchmarks from which we extracted call graphs were reduced to a suite with either 19, 25, or 27 microbenchmarks.

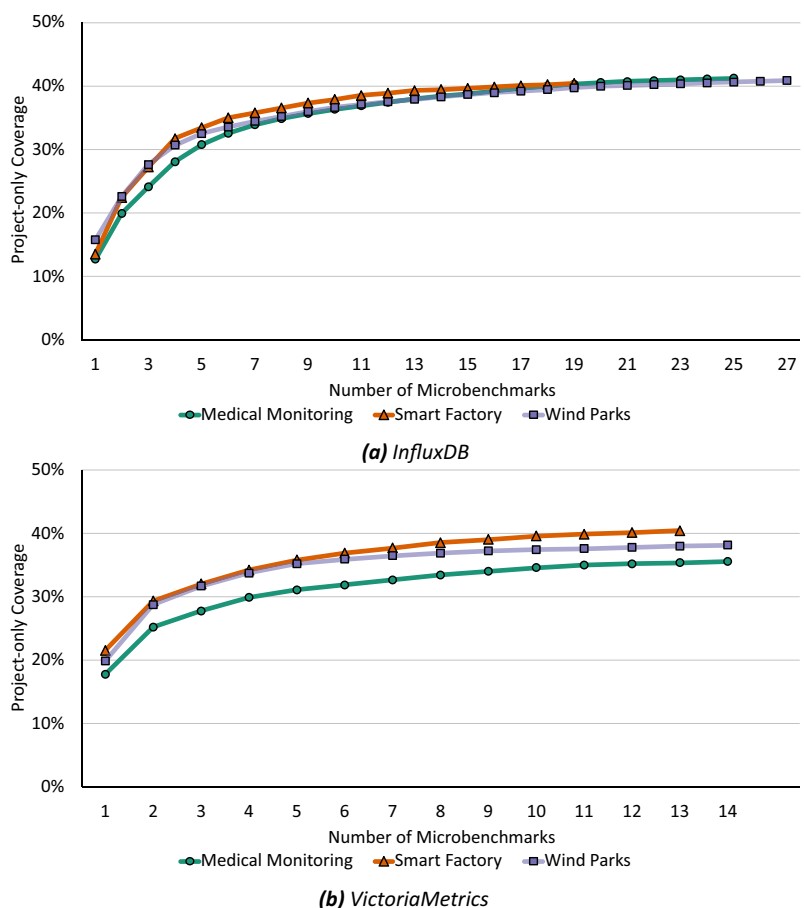

**Figure 6** Already the first four microbenchmarks selecting by Algorithm 1 cover 28% to 31% for *InfluxDB* (A) and 29% to 34% for *VictoriaMetrics* (B) of the respective application benchmark's call graph.

In general, we find similar results for *VictoriaMetrics* (Fig. 6B). Already the first microbenchmark selected by our algorithm covers at least 17% of the application benchmark call graph in each scenario. For *VictoriaMetrics*, the first four selected microbenchmarks also have similar coverage sets, there is only a small difference in the parametrization of one chosen microbenchmark. In total, these first four microbenchmarks cover 29% to 34% of the application benchmark call graph, depending on the scenario, and there is a maximum possible coverage between 35% and 40% when using the full existing microbenchmark suite. Again, the first four microbenchmarks are therefore particularly effective and already cover a large part of the application benchmark call graph. Moreover, even with the complete minimization and the maximum possible coverage, our algorithm significantly reduces the number of microbenchmarks: from 62 microbenchmarks down to 13 or 14 microbenchmarks, depending on the concrete application scenario.

Since each microbenchmark takes on average about the same amount of time (see "Microbenchmarks"), our minimal suite results in a significant time saving when running the microbenchmark suite. For *InfluxDB* it would take only about 10% of the original time

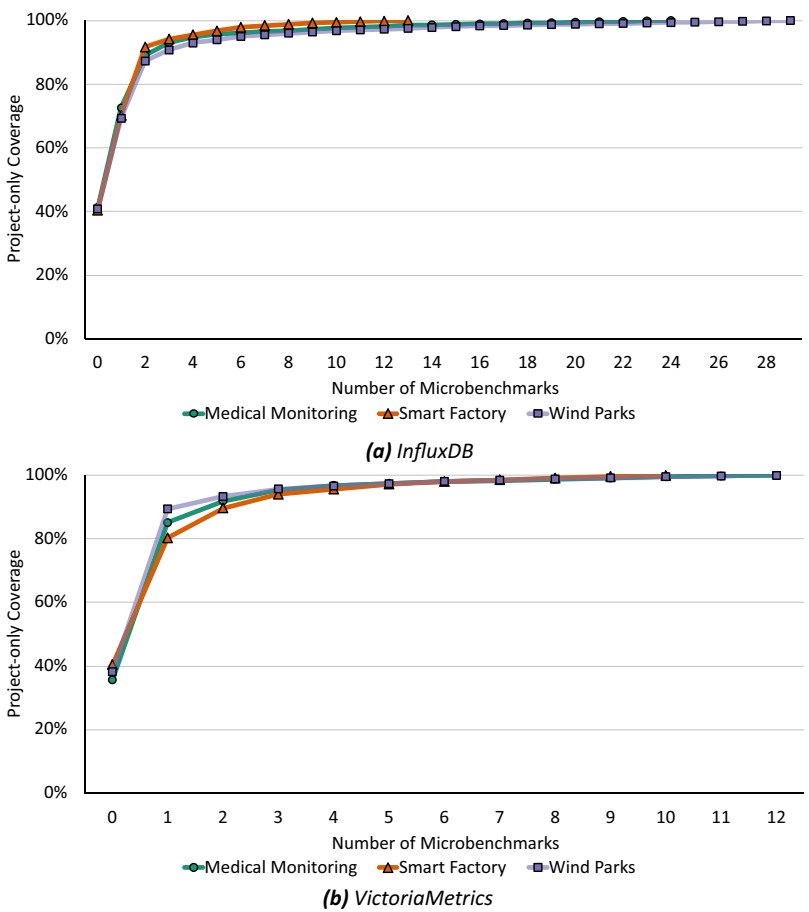

**Figure 7 Already microbenchmarks of the first three recommended functions could increase the project-only coverage up to 90% to 94% for *InfluxDB* (A) and 94% to 95% for *VictoriaMetrics* (B).**

and for *VictoriaMetrics* about 23% respectively. On the other hand, these drastic reductions also mean that many microbenchmarks in both projects evaluate the same functions. This can be useful under certain circumstances, e.g., if there is a performance degradation detected using the minimal benchmark suite and developers need to find the exact cause. However, given our goal of finding a minimal set of microbenchmarks to use as smoke test in a CI/CD pipeline, these redundant microbenchmarks present an opportunity to drastically reduce the execution time without much loss of information.

## Recommending additional microbenchmark targets

Our second optimization, the recommendation, starts with the minimal microbenchmark suite from above and subsequently recommends functions to increase the coverage of the microbenchmark suite and application benchmark following Algorithm 2. Figure 7 shows this step-by-step recommendation of functions starting with the current coverage up to a 100% relevant microbenchmark suite.

For *InfluxDB* (Fig. 7A), a microbenchmark for the first recommended function would increase the coverage by 28% to 31% depending on the application scenario and the first

three recommended functions are identical for all scenarios: (i) *executeQuery* runs a query against the database and returns the results, (ii) *ServeHTTP* responds to HTTP requests, and (iii) *storeStatistics* writes statistics into the database. If each of these functions were evaluated by a microbenchmark in the same way as the application benchmark, i.e., resulting in the same calls of downstream functions and the same call graph, there would already be a total coverage of 90% to 94%. To achieve a 100% match, additional 10 to 26 functions must be microbenchmarked, depending on the application scenario and always under the assumption that the microbenchmark will call the function in the same way as the application benchmark does.

In general, we find similar results for *VictoriaMetrics* (Fig. 7B). Already a microbenchmark for the first recommended function would increase the coverage by 39% to 51% and microbenchmarking the first three recommended functions would increase the coverage up to a total of 94% to 95%. Again, these three functions are recommended in all scenarios, only the ordering is different. All first recommended functions are anonymous functions, respectively (i) an HTTP handler function, (ii) a merging function, and (iii) a result-related function. To achieve 100% project-only coverage, 10 to 14 additional functions would have to be microbenchmarked depending on the application scenario.

In summary, our results show that the microbenchmark suite can be made much more relevant to actual practice and usage with only a few additional microbenchmarks for key functions. In most cases, however, it will not be possible to convert the recommendations directly into suitable microbenchmarks (we discuss this point in the next section). Nevertheless, we see these recommendations as a valid starting point for more thorough analysis.

## DISCUSSION

We propose an automated approach to analyze and improve microbenchmark suites. It can be applied to all application systems that allow the profiling of function calls and the subsequent creation of a call graph. This is particularly easy for projects written in the Go programming language as this functionality is part of the Go environment. Furthermore, our approach is beneficial for projects with a large code base where manual analysis would be too complex and costly. In total, we propose three methods for analyzing and optimizing existing microbenchmark suites but can also provide guidance for creating new ones. Nevertheless, every method has its limits and should not be applied blindly.

Assuming that the application benchmark reflects a real production system or simulates a realistic situation, the resulting call graphs will reflect this perfectly. Unfortunately, this is not always the case, because the design and implementation of a sound and relevant application benchmark has its own challenges and obstacles which we will not address here (*Bermbach, Wittern & Tai, 2017*). Nevertheless, a well-designed application benchmark is capable of simulating different scenarios in realistic environments in order to identify weak points and to highlight strengths. Ultimately, however, for the discussion that follows, we must always be aware that the application benchmark will never be a perfect

representation of real workloads. Trace-based workloads (*Bermbach et al., 2017*) can help to introduce more realism.

**Considering only function calls is imperfect but sufficient:** Our approach relies on identifying the coverage of nodes in call graphs and thus on the coverage of function calls. Additional criteria such as path coverage, block coverage, line coverage, or the frequency of function executions are not considered and subject to future research. We deliberately chose this simple yet effective method of coverage measurement: (1) Applying detailed coverage metrics such as line coverage would deepen the analysis and check that every code line called by the application benchmark is at least once called by a microbenchmark. However, if the different paths in a function source code are relevant for production and do not only catch corner cases, they should be also considered in the application benchmark and microbenchmark workload (e.g., if the internal function calls in the Medical Monitoring scenario would differ for female and male patients, the respective benchmark workload should represent female and male patients with the same frequency as in production). (2) As our current implementation relies on sampling, the probability that a function that is called only once or twice during the entire application benchmark or microbenchmark is called at the exact time a sample is taken is extremely low. Thus, the respective call graphs will usually only include practically relevant functions. (3) We assume that all benchmarks adhere to benchmarking best practices. This includes both the application benchmark which covers all relevant aspects and the individual microbenchmarks which each focus on individual aspects. This implies that if there is an important function, this function will usually be covered by multiple microbenchmarks which each generate a unique call graph with individual function calls and which therefore will all be included into the optimized microbenchmark suite. Thus, there will still usually be multiple microbenchmarks which evaluate important functions. (4) Both base algorithms (*Chen & Lau, 1998*; *Rothermel et al., 1999*) are standard algorithms and have recently been shown to work well with modern software systems (e.g., *Luo et al., 2018*). We therefore assume that a relevant benchmark workload will generate a representative call graph and argue that a more detailed analysis of the call graph would not improve our approach significantly. The same applies to the microbenchmarks and their coverage sets with the application benchmark where our approach will only work if the microbenchmark suite generates representative function invocations. Overall, the optimized suite serves as simple and fast heuristic for detecting performance issues in a pre-production stage but it is—by definition—not capable of detecting all problems: there will be false positives and negatives. In practice, we would therefore suggest to use the microbenchmark-based heuristic with every commit whereas the application benchmark will be run periodically; how often is subject to future research.

**The sampling rate affects the accuracy of the call graphs:** The generation of the call graphs in our evaluation is based on statistical sampling of stack frames at specified intervals. Afterwards, the collected data is combined into the call graph. However, this carries the risk that, if the experiment is not run long enough, important calls might not be registered and thus will not appear in the call graph. The required duration depends on the software project and on the sampling rate, i.e., at which frequency samples are taken. To

account for this, we chose frequent sampling combined with a long benchmark duration in our experiments which makes it unlikely that we have missed relevant function calls.

**Our approach is transferable to other applications and domains:** We have evaluated our approach with two TSDBs written in the Go programming language, but we see no major barriers to implementing our approach for applications written in other programming languages. There are several profiling tools for other programming languages, e.g., for Java or Python, so this approach is not limited to the Go programming language and is applicable to almost all software projects. Moreover, there are various other application domains where benchmarking can be applied which we also discuss in "Related Work". In this work, we primarily intend to present the approach and its resulting opportunities, e.g., for CI/CD pipelines. The transfer to other application domains and programming languages is subject to future research.

**The practical relevance of a microbenchmark suite can be quantified quickly and accurately:** Our approach can be used to determine and quantify the practical relevance of a microbenchmark suite based on a large baseline call graph (e.g., an application benchmark) and many smaller call graphs from the execution of the microbenchmark suite. On one hand, this allows us to determine and quantify the practical relevance of the current microbenchmark suite with respect to the actual usage: in our evaluation of two different TSDBs, we found that this is 40% for both databases. On the other hand, this means that 60% of the required code parts for the daily business are not covered by any microbenchmark, which highlights the need for additional microbenchmarks to detect and ultimately prevent performance problems in both study objects. It is important to note that the algorithm only includes identical nodes in the respective graphs; edges, i.e., which function calls which other function, are not considered here. This might lead to an effectively lower coverage if our algorithms selects a microbenchmark that only measures corner cases. To address this, it may be necessary to manually remove all microbenchmarks that do not adhere to benchmarking best practices before running our algorithm. In summary, we offer a quick way to approximate coverage and practical relevance of a microbenchmark suite in and for realistic scenarios.

**A minimal microbenchmark suite with reduced redundancies can be used as performance smoke test:** Our first optimization to an existing microbenchmark suite, Algorithm 1, aims to find a minimal set of microbenchmarks which already cover a large part of an application benchmark, again based on the nodes in existing call graphs. Our evaluation has shown that a very small number of microbenchmarks is sufficient to cover a large part of the potential maximum coverage for both study objects. Furthermore, it has also shown that the number of microbenchmarks in a suite can still be significantly reduced, even if we want to achieve the maximum possible coverage. Translated into execution time, this removal of redundancies corresponds to savings of up to 90% in our scenarios, which offers a number of benefits for benchmarking in CI/CD pipelines. A minimal microbenchmark suite could show developers a rough performance impact of their current changes. This enables developers to run a quick performance test on each commit, or to quickly evaluate a new version before starting a more complex and cost-intensive application benchmark. In this setup, the application benchmark remains the

gold standard to detect all performance problems while the less accurate optimized microbenchmark suite is a fast and easy-to-use performance check. Finally, it is important to note that the intention of our approach is not to remove "unnecessary" microbenchmarks entirely but rather to define a new microbenchmark suite as a subset of the existing one which serves as a proxy to benchmarking the performance of the SUT. Although our evaluation also revealed that many microbenchmarks benchmark the same code and are therefore redundant, this redundancy is frequently desirable in other contexts (e.g., for detailed error analysis).

**The recommendations can not always be directly used:** Our second optimization, Algorithm 2, recommends functions which should be microbenchmarked in order to cover a large additional part of realistic application flow in the SUT. Our evaluation with two open-source TSDBs has shown that this is indeed possible and that already with a small number of additional microbenchmarks a large part of the application benchmark call graph could be covered. However, our evaluation also suggests that these microbenchmarks are not always easy to implement, as the recommended functions are often very generic and abstract. Our recommendation should therefore mostly be seen as an initial point for further manual investigation by expert application developers. Using their domain knowledge, they can estimate which (sub)functions are called and what their distribution/ratio actually is. Furthermore, the application benchmark's call graph can also support this analysis as it offers insights into the frequency of invocation for all covered functions.

## RELATED WORK

Software performance engineering traditionally revolves around two general flavors: model-based and measurement-based. The context of our study falls into measurement-based software performance engineering, which deals with measuring certain performance metrics, e.g., latency, throughput, memory, or I/O, over time. Research on application benchmarking and microbenchmarking topics are related to our study, in particular for reducing their execution frequency or generating new microbenchmarks.

### Application benchmarking

Related work in this area deals with the requirements for benchmarks in general, application-specific characteristics, and more effective benchmark execution. Furthermore, contributors expand on the analysis of problems and examine the influence of environmental factors on the benchmark run in more detail.

One of the earliest publications addressed general challenges such as testing objectives, workload characterization, and requirements (*Weyuker & Vokolos, 2000*). These aspects were then refined and adapted to present needs and conditions in an ongoing process (e.g., *Huppler, 2009*; *Bermbach et al., 2017*; *Bermbach, Wittern & Tai, 2017*; *Folkerts et al., 2013*).

Current work focuses on application-specific benchmarks. To name a few, there are benchmarks which evaluate database or storage systems (e.g., *Bermbach et al., 2014*; *Cooper et al., 2010*; *Bermbach et al., 2017*; *Kuhlenkamp, Klems & Röss, 2014*; *Müller et al., 2014*; *Pallas et al., 2017*; *Pallas, Günther & Bermbach, 2017*; *Pelkonen et al., 2015*; *Difallah*

*et al., 2013*), benchmark microservices (*Villamizar et al., 2015*; *Grambow et al., 2020*; *Grambow, Wittern & Bermbach, 2020*; *Ueda, Nakaike & Ohara, 2016*; *Do et al., 2017*), determine the quality of web APIs (*Bermbach & Wittern, 2016*, *2020*), specifically tackle web sites (*Menascé, 2002*), or evaluate other large-scale software systems (e.g., *Jiang & Hassan, 2015*; *Hasenburg et al., 2020*; *Hasenburg & Bermbach, 2020*). Our approach can use all of these application benchmarks as a baseline. As long as a call graph can be generated from respective SUT during the benchmark run, this graph can serve as input for our approach.

Other approaches aim to reduce the execution time for application benchmarks: *AlGhamdi et al. (2016*, *2020)* proposed to stop the benchmark run when the system reaches a repetitive performance state, and *He et al. (2019)* devised a statistical approach based on kernel density estimation to stop once a benchmark is unlikely to produce a different result with more repetitions. Such approaches can only be combined with our analysis and optimization under certain conditions. The main aspect here are rarely called functions which might never be called if the benchmark run is terminated early. If the determination of the call graph is based on sampling, as in our evaluation, the results could be incomplete because relevant calls were not detected.

Many studies and approaches address the factors and conditions in cloud environments during benchmarks (*Binnig et al., 2009*; *Difallah et al., 2013*; *Folkerts et al., 2013*; *Kuhlenkamp, Klems & Röss, 2014*; *Silva et al., 2013*; *Rabl et al., 2010*; *Schad, Dittrich & Quiané-Ruiz, 2010*; *Iosup, Yigitbasi & Epema, 2011*; *Leitner & Cito, 2016*; *Laaber, Scheuner & Leitner, 2019*; *Uta et al., 2020*; *Abedi & Brecht, 2017*; *Bermbach, 2017*). These studies and approaches are relevant for the application of our approach. If the variance in the test environment is known and can be reduced to a minimum, this supports the application engineers in deciding at what time and to which extent which benchmark type should be executed.

Finally, there are several studies that aim to identify (the root cause of) performance regressions (*Nguyen et al., 2014*; *Foo et al., 2015*; *Daly et al., 2020*; *Grambow, Lehmann & Bermbach, 2019*; *Waller, Ehmke & Hasselbring, 2015*) or examine the influence of environment factors on the system under test, such as the usage of Docker (*Grambow et al., 2019*). Here, the first mentioned approaches can be combined with our approach very well. If a performance problem is not detected although the microbenchmark suite is optimized, the mentioned approaches can be used in the secondary application benchmark to support developers. Regarding the environmental parameters, these must be taken into account to achieve a reliable and relevant result. If the execution of functions depends on specific environmental factors which differ between test and production environment, this can falsify the outcome.

## Microbenchmarking

The second form of benchmarking that is subject of this study is microbenchmarking, which has only recently gained more traction from research. *Leitner & Bezemer (2017)* and *Stefan et al. (2017)* empirically studied how microbenchmarks—sometimes also referred to as performance unit tests – are used in open-source Java projects and found that

adoption is still limited. Others focused on creating performance-awareness through documentation (*Horký et al., 2015*) and removing the need for statistical knowledge through simple hypothesis-style, logical annotations (*Bulej et al., 2012*, *2017*). *Chen & Shang (2017)* characterize code changes that introduce performance regressions and show that microbenchmarks are sensitive to performance changes. *Damasceno Costa et al. (2019)* study bad practices and anti-patterns in microbenchmark implementations. All these studies are complementary to ours as they focus on different aspects of microbenchmarking that is neither related to time reduction nor recommending functions as benchmark targets.

*Laaber & Leitner (2018)* are the first to study microbenchmarks written in Go and apply a mutation-testing-inspired technique to dynamically assess redundant benchmarks. Their idea is similar to ours: we use static call graphs to compute the microbenchmark coverage of application benchmark calls, whereas they compute redundancies between microbenchmarks of the same suite. *Ding, Chen & Shang (2020)* study the usability of functional unit tests for performance testing and build a machine learning model to classify whether a unit test lends itself to performance testing. Our redundancy removal approach could augment their approach by filtering out unit tests (for performance) that lie on the hot path of an application benchmark.

To reduce the overall microbenchmark suite execution time, one might execute the microbenchmarks in parallel on cloud infrastructure. Recent work studied how and to which degree such an unreliable environment can be used (*Laaber, Scheuner & Leitner, 2019*; *Bulej et al., 2020*). Similar to *He et al. (2019)* but for microbenchmarks, *Laaber et al. (2020)* introduced dynamic reconfiguration to stop the execution when their result is stable in order to reduce execution time. Our approach to remove redundancies is an alternative approach to reduce microbenchmark suite execution time.

Another large body of research is performance regression testing, which utilizes microbenchmarks between two commits to decide whether and what to test for performance. *Huang et al. (2014)* and *Sandoval Alcocer, Bergel & Valente (2016*, *2020)* utilize models to assess whether a code commit introduces a regression to select versions that should be tested for performance. *de Oliveira et al. (2017)* and *Alshoaibi et al. (2019)* decide based on source code indicators which microbenchmarks to execute on every commit. *Mostafa, Wang & Xie (2017)* rearrange microbenchmarks to execute the ones earlier that are more likely to expose performance changes. These studies focus on reducing the time of performance testing or focusing on the relevant microbenchmarks/ commits, which is similar in concept to our motivation. Our study, however, utilizes different granularity levels of performance tests, i.e., application benchmarks and microbenchmarks, to inform which microbenchmarks are more or less relevant.

Finally, synthesizing microbenchmarks could be a way to increase coverage of important parts of an application. These could, for instance, be identified by an application benchmark. SpeedGun generates microbenchmarks for concurrent classes to expose concurrency-related performance bugs (*Pradel, Huggler & Gross, 2014*); and AutoJMH randomly generates microbenchmark workloads based on forward slicing and control flow graphs (*Rodriguez-Cancio, Combemale & Baudry, 2016*). Both approaches are highly

related to our paper as they propose solutions for not yet existing benchmarks. However, both require as input a class or a segment that shall be performance tested. Our recommendation algorithm could provide this input.

## CONCLUSION

Performance problems of an application should ideally be detected as soon as they occur. Unfortunately, it is often not possible to verify the performance of every source code modification by a complete application benchmark for time and cost reasons. Alternatively, much faster and less complex microbenchmarks of individual functions can be used to evaluate the performance of an application. However, their results are often less meaningful because they do not cover all parts of the source code that are relevant in production.

In this paper, we determine, quantify, and improve this practical relevance of microbenchmark suites based on the call graphs generated in the application during the two benchmark types and suggest how the microbenchmark suite can be designed and used more effectively and efficiently. The central idea of our approach is that all functions of the source code that are called during an application benchmark are relevant for production use and should therefore be covered by the faster and more lightweight microbenchmarks as well. To this end, we determine and quantify the coverage of common function calls between both benchmark types, suggest two methods of optimization, and illustrate how these can be leveraged to improve build pipelines: (1) by removing redundancies in the microbenchmark suite, which reduces the total runtime of the suite significantly; and (2) by recommending relevant target functions which are not covered by microbenchmarks yet to increase the practical relevance.

Our evaluation on two time series database systems shows that the number of microbenchmarks can be significantly reduced (up to 90%) while maintaining the same coverage level and that the practical relevance of a microbenchmark suite can be increased from around 40% to 100% with only a few additional microbenchmarks for both investigated software projects. This opens up a variety of application scenarios for CI/CD pipelines, e.g., the optimized microbenchmark suite might scan the application for performance problems after every code modification or commit while running the more complex application benchmark only for major releases.

In future work, we plan to investigate whether such a build pipeline is capable of detecting and catching performance problems at an early stage. Furthermore, we want to examine if a more detailed analysis of our coverage criteria on path or line level of the source code is feasible and beneficial. Even though there are still some limitations, we think that our automated approach is very useful to support larger software projects in detecting performance problems effectively, in a cost-efficient way, and at an early stage.

### Funding

Philipp Leitner received financial support from the Swedish Research Council VR under grant number 2018-04127 (Developer-Targeted Performance Engineering for Immersed

Release and Software Engineers). The funders had no role in study design, data collection and analysis, decision to publish, or preparation of the manuscript.

## Grant Disclosures
The following grant information was disclosed by the authors:
Swedish Research Council VR: 2018-04127.

## Competing Interests
Philipp Leitner is an Academic Editor for PeerJ Computer Science.

## Author Contributions
- Martin Grambow conceived and designed the experiments, performed the experiments, analyzed the data, performed the computation work, prepared figures and/or tables, authored or reviewed drafts of the paper, and approved the final draft.
- Christoph Laaber conceived and designed the experiments, performed the experiments, analyzed the data, performed the computation work, prepared figures and/or tables, authored or reviewed drafts of the paper, and approved the final draft.
- Philipp Leitner conceived and designed the experiments, prepared figures and/or tables, authored or reviewed drafts of the paper, and approved the final draft.
- David Bermbach conceived and designed the experiments, prepared figures and/or tables, authored or reviewed drafts of the paper, and approved the final draft.

## Data Availability
Code and results are available in the Supplemental File.
The adjusted version of YCSB-TS is available at GitHub: https://github.com/martingrambow/YCSB-TS.

## Supplemental Information
Supplemental information for this article can be found online at http://dx.doi.org/10.7717/peerj-cs.548#supplemental-information.

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
