# Peer review of "Using application benchmark call graphs to quantify and improve the practical relevance of microbenchmark suites"

_PeerJ Computer Science, doi:10.7717/peerj-cs.548_

## Round 0.1 · original submission · Major Revisions

Dear authors, overall the paper is well written. However, certain technical aspects need revisions (as mentioned in the below reviews). I hope that you will address (in a comprehensive way) all the concerns raised by the reviewers. Thanks.

·

Basic reporting

The submission describes a method by which to minimise a suite of microbenchmarks for use within a CI pipeline, where quick feedback is desirable. Given a reference "application benchmark" and a suite of microbenchmarks, the method uses coverage metrics to boil down the microbenchmark suite into a (hopefully) representative set of bare-minimum microbenchmarks.

In addition, a method is also proposed for bulking up a too-small microbenchmark suite, by pointing out areas of code which are not yet covered.

The submission is well-written and the spelling and grammar is excellent. Referencing seems sufficient and I have no issues with the structure of the article.

The approach is well-explained and I feel that I have a decent grasp of what the authors are proposing.

Experimental design

Unfortunately, I don't think the design of the experiment is quite right.

It appears that the approach is underpinned by the assumption that microbenchmarks which cover the same functions are equal. From experience, it is not only code coverage that characterises a benchmark, but also how frequently certain parts of code are executed.

To frame this in the context of the paper's approach, suppose that a reference application benchmark AB spends 90% of its time in a tight loop calling a function F1. That means that benchmarking measurements of AB are very much coupled with the performance of F1. Inefficiencies in this function will be amplified because the function is called so frequently. If AB is truly a proxy for the production environment, then F1 had better be really well optimised!

Now suppose that there's a microbenchmark MB1 that executes (covers) exactly the same set of functions as AB, but which only calls F1 very infrequently. Suppose that in MB1 only 1% of runtime is spent in F1. The proposed method would select MB1 because, according to the paper's definition of "practical relevance", this is the ideal microbenchmark: the coverage of both the AB and MB1 are identical. Yet performance measurements of MB1 are highly unlikely to discover inefficiencies in F1, since 99% of runtime is spent elsewhere and any performance irregularities caused by F1 are therefore lost in the noise. Thus MB1 is a poor proxy for AB and if F1 did have performance issues, MB1 would fail to find them, whereas running AB would find them immediately.

The experiments conducted then measure the ability for the proposed approach to minmimise the microbenchmark suite into one with similar code coverage as the application benchmark. I'd argue that instead the evaluation should check whether the reduced benchmark suite has similar ability to detect the same performance problems as the application benchmark. At the very least, I think the evaluation should test whether code coverage is a good proxy for this (I doubt it is, but I'm willing to be proven wrong).

To fix this approach properly, the definition of practical relevance would need to be revised to take "hotness" (frequency of execution) of functions into account. Perhaps function nodes in the call graph need some kind of weighting applied.

Validity of the findings

I think the validity of the findings is limited due to the problems outlined in the previous section.

Additional comments

How do you deal with non-determinism in benchmarks, where different parts of the CFG are executed in subsequent executions? Presumably you'd union the CFGs of some 30 runs?

In Section 3.1 I wasn't sure where 6/16 comes from. Should it be 6/19?

"our proof of concept simply selects a random coverage set in case of a tie" -- This introduces non-determinism into your algorithm. Should you be repeating your experiments (say) 30 times and reporting some measure of variation then? You also mention sampling to get the CFG, which is another source of non-determinism.

"covers the application benchmark to the same extend" -> "... to the same extent"

"coverage of all covered function" -> "coverage of all covered functions" or even better "coverage of all executed functions".

Algorithm 2: "{ A | a \in A \wedge ...}" should that be "{ a | a \in A ...}" (lower case 'a' in the set comprehension)?

·

Basic reporting

Following are a few ambiguous statements that should be reviewed.

"Depending on the application scenario, the 288 microbenchmarks from which we extracted call graphs were reduced to either 19, 25, or 27."

The context of private and business customers is not clear.
"e.g., if a function calls different other functions depending on whether or not a value corresponds to a business customer, the respective benchmark workload should represent private and business customers in the same distribution..."

"Thus, the determined coverage corresponds to a maximum value, and the actual value might be smaller. "

"Finally, synthesizing microbenchmarks could be a way to increase coverage of important parts of an application, as, for example, identified by an application benchmark."

Experimental design

One of the objectives of the article is to remove the redundancies in micro-benchmark suites. Authors should address the payload aspect because behavior of the redundant benchmark may vary depending on the inherit nature of the payload.

Call graph related approaches are usually used in software testing domain. How can the definition of SUT be satisfied if one function is called only once in the micro benchmark? Authors have agreed that benchmark cannot be a perfect representative of a real application but in this case, a system cannot be characterized as 'under stress' if a benchmark function is called only once. The authors do claim in the last section that benchmark relevance can be increased up to 100% that seems contradictory to previous statement.

The paper has built the proposed technique using the two algorithms of Chen and Lau (1998) and Rothermel et al. (1999) that overlap with software testing domain. Considerable work has been done in the software testing domain to ensure comprehensive coverage. The authors should provide arguments that these relatively old techniques can achieve desired goals in all kinds of applications.

Why is there a need for calling SortSet(C) in each iteration when C does no change during the later steps in Algo 1?

How n will be determined representing the number of benchmarks to be recommended used in Algo 2?Can it not be determined heuristically?

Figure 2 presents an example with 2 entry points. How efficient will be the proposed technique in case the redundancy is not at application level? Figure 2 shows 19 nodes and its is not clear that how overall coverage comes out to be 6/16?

Validity of the findings

Various results have been presented that are related with the RQ2 however the the crucial aspect of "how to improve execution efficiency" is not obvious from the results and discussion.

The authors have suggested that the proposed approach is valid for application that allow profiling in the context of call graphs. Is there any estimate for the percentage of applications that have this characteristic?

Authors should explore the reasons that all 3 applications behave very similarly in the context of Figure 7. It can be argued using Figure 7 that there is no need to include more than 5-6 benchmarks. Is this result dependent on the ordering of the inclusion of the benchmarks?

If 40% coverage is provided by the micro-benchmarks used in this experiment, how satisfied is the community that have been using these micro-benchmarks? Were they getting any contradicting results? If yes, are their any efforts to replace these benchmarks with better micro-benchmarks?

---

## Round 0.2 · accepted · Accept

Congratulations, the revised version of the manuscript is recommended for publication.

Reviewer 3 ·

Basic reporting

OK

Experimental design

OK

Validity of the findings

OK

Additional comments

The authors has fulfilled all previous comments